# Generated Image Editing Method Based on Global-Local Jacobi Disentanglement for Machine Learning

**DOI:** 10.3390/s23041815

**Published:** 2023-02-06

**Authors:** Jianlong Zhang, Xincheng Yu, Bin Wang, Chen Chen

**Affiliations:** 1School of Electronic Engineering, Xidian University, Xi’an 710071, China; 2State Key Laboratory of Integrated Services Networks, Xidian University, Xi’an 710071, China

**Keywords:** StyleGAN2, unsupervised methods, image editing, weight matrix eigen decomposition, Jacobi orthogonal regularization

## Abstract

Accurate semantic editing of the generated images is extremely important for machine learning and sample enhancement of big data. Aiming at the problem of semantic entanglement in generated image latent space of the StyleGAN2 network, we proposed a generated image editing method based on global-local Jacobi disentanglement. In terms of global disentanglement, we extract the weight matrix of the style layer in the pre-trained StyleGAN2 network; obtain the semantic attribute direction vector by using the weight matrix eigen decomposition method; finally, utilize this direction vector as the initialization vector for the Jacobi orthogonal regularization search algorithm. Our method improves the speed of the Jacobi orthogonal regularization search algorithm with the proportion of effective semantic attribute editing directions. In terms of local disentanglement, we design a local contrast regularized loss function to relax the semantic association local area and non-local area and utilize the Jacobi orthogonal regularization search algorithm to obtain a more accurate semantic attribute editing direction based on the local area prior MASK. The experimental results show that the proposed method achieves SOTA in semantic attribute disentangled metrics and can discover more accurate editing directions compared with the mainstream unsupervised generated image editing methods.

## 1. Introduction

Machine learning [1], big data [2,3], and the Internet of Things (IoT) [4] have infiltrated every aspect of people's lives. Generative adversarial networks (GANs) [5] have been viewed as the most interesting ideas of artificial intelligence in the last decade. Being the most promising generative models, GANs are widely used in areas, such as image generation, text synthesis, and style transformation. In the field of image generation, GAN networks, e.g., StyleGAN [6], PGGAN [7], StyleGAN2 [8], and BigGAN [9], have been able to generate realistic images with high resolution as the network layers deepen and the scale increases. However, in many practical applications, the problem faced by GAN networks is the uncontrollability of generation, i.e., the randomness of sampling of latent space, and their generated results cannot follow human wishes. For example, it can generate realistic face images, but we cannot control the generated characters, e.g., male or female, long or short hair, etc. In application scenarios, e.g., portrait editing, modeling, animation character design, criminal investigation, and big data generation and enhancement, not only the quality of the generated images should be ensured, but also editability is required. Figure 1 shows an example of hairstyle, age, pose, and gender editing on the face image generated by the StyleGAN2 network. The controllable editing of faces makes it possible to apply to GAN for portrait depiction and movie art production. Applications such as the generation and editing of portraits of anime characters and criminal suspects in criminal investigation no longer require a painter to draw images manually, saving labor and material resources. Regeneration and re-editing of the image data acquired by the sensors can greatly expand the learning sample.

With further research on the mechanism of GAN, editing methods to control GAN-generated images have emerged, and these methods can be mainly divided into supervised and unsupervised methods.

Supervised methods require semantic labels for training dataset images. CGAN [10] trains GAN networks to generate images by the training set with given semantic labels; it could generate images according to the established semantic information of the labels, but cannot realize the continuous editing of images in certain semantics. GAN-Control [11] solves the problem by quantifying the semantic properties of the labels with a large number of semantic labels. InterFaceGAN [12] samples the latent space Z; employs a support vector machine (SVM) classifier to determine the semantic direction of the latent space; realizes precise image editing along the semantic attribute direction, however, its training classifier requires a large number of binary classification samples. In the field of supervised local image editing, Spatially Control GAN [13] achieves local area feature blending of two images by blending the latent spaces corresponding to each semantic attribute, but the method suffers from the incongruence of the generated images after blending. SemanticStyleGAN [14] achieves local image editing by assigning specific masks to the generated images and corresponding the masks to the latent spaces, however, the addition of the semantic segmentation model makes the model too large and bloated, which is not conducive to lightweight deployment. Spatial Attention GAN [15] achieves local editing of images by introducing a spatial attention mechanism and classifying each attribute.

Supervised methods can edit images accurately but it is difficult to obtain semantic labels, so unsupervised methods have gradually become a new research trend. The unsupervised methods do not require external semantic attribute labels; they mainly utilize the potential semantic directions in the latent space of the GAN [16]; the parameters are trained in the generator network [17]. PCAGAN [18] utilizes principal component analysis (PCA) [19] to determine the main feature direction of controlled editing in the latent space of the GAN; find the main direction of image semantic attribute; however, its result depends on the number of sampled points in the latent space and the entanglement between each semantic attribute is serious. SefaGAN [20] proposed that the main semantic attribute directions of the generated images are the eigenvectors of the fully connected layer weight matrix in the pre-trained model of the StyleGAN2 network. It can obtain interpretable image semantic attribute directions, but its equivalent model is not complete. It equates the mapping function in the generator network with the fully connected layer of the style layer; while omitting the mapping function of other layers in the network, which results in the actual editing process that still has semantic attribute entanglement [21]. As shown in Figure 2, when editing along with the gender semantic attribute direction, the faces in red boxes 1 and 2 have added glasses, and the hair color has changed. STGAN-WO [22] controls image generation at two scales separately in the latent space and achieves disentanglement of image structure from texture semantic attribute using weight orthogonal regularization, but its regularization method limits the weight space in the network; consequently, reduces the quality of the generated images. Chen et al. [23,24]. achieves good results using spatial features of deep convolutional neural networks for image feature detection. The multi-scale hierarchical disentangled image editing method [25] achieves more accurate image editing on the multi-scale feature space, in which the weight matrix at different resolution layers of the StyleGAN2 generator network is decomposed and the semantic attributes between different resolution layers are disentangled by using Schmitt orthogonal decomposition, however, it does not fundamentally change the limitations of the weight matrix eigen decomposition method, but only alleviates the semantic entangled phenomenon at multiple scales. LowRankGAN [26] obtains the Jacobi matrix by calculating the partial derivatives of the generated feature map of the GAN with each vector element of the input latent space; obtain the image editing directions by employing low-rank approximation and SVD decomposition to the Jacobi matrix. It achieves better editing results than the previous methods, but the calculation of the Jacobi matrix is time-consuming and memory-intensive. Hessian Penalty GAN [27] adopts the idea of the Jacobian matrix to restrict each dimension of the input latent space by orthogonal regularization, the method achieves different semantic attributes of the generated image controlled by different dimensions of the latent space, but the restriction of different dimensions to control a single semantics is too strict, and the editing does not work well on models with large hidden space dimensions such as StyleGAN. OroJaRGAN [28] relaxes the orthogonal regularization constraints on the basis of Hessian Penalty GAN, the method achieves good results on large GAN models.

Although a good disentangled result is achieved in the generated image editing, the Jacobi matrix is highly complex because of the image’s high resolution and the latent space’s numerous dimensions. Currently, there is little research on how to accelerate algorithms related to Jacobi computing. However, there are still some methods to optimize the semantic direction search process for Jacobi disentanglement, such as LowRankGAN and OroJaRGAN. LowRankGAN reduces the complexity of the Jacobi matrix using low-rank approximation, which reduces the difficulty of semantic direction decomposition, however, low-rank approximation increases the overhead of the algorithm and still requires the direct computation of the Jacobi matrix. OroJaRGAN utilizes Hutchinson’s estimator and Rademacher vectors to avoid the direct computation of Jacobi matrices and uses the difference derivative method to reduce the complexity of the computation, however, the search with random initialization makes the time cost is still high and there are a large number of semantic directions for ineffective search.

The unsupervised image editing method based on the StyleGAN2 network can obtain interpretable semantic attribute direction, but there are the following problems: (1). For high-resolution and semantically complex generated images, serious semantic attribute entangled phenomenon still remains. (2). The weight matrix eigendecomposition method only extracts the style layer weights in the StyleGAN2 network and ignores the mapping function of other layers, which is difficult to obtain accurate semantic directions. (3). Jacobi orthogonal regularization search algorithm can obtain the global accurate image editing directions, but the search process is time-consuming besides ineffective editing directions. (4). Local semantic attributes that are highly entangled with global features cannot be disentangled from the global image, and it is difficult for ordinary methods to discover more accurate local semantic directions.

To address the above issues, we proposed a generated image editing method based on global-local Jacobi disentanglement. The main contributions of the work are as follows.
A new global Jacobi orthogonal regularization search semantic direction set initialization method is designed, using the semantic direction vector of the weight matrix eigendecomposition as the initial vector, which improves the search speed and reduces the proportion of ineffective search directions.A local Jacobi disentangled method is proposed to discover more accurate image editing directions by limiting the search area and designing a contrast regularized loss function.Experiments on the datasets FFHQ and LSUNCat show that our method achieves optimality in semantic attribute disentangled metrics compared to existing unsupervised generated image editing methods, and is also able to discover more accurate image editing directions.

The rest of this paper is organized as follows. In Section 2, we expound on the principles and implementation of the generated image editing method proposed in this paper. Section 3 provides experimental evaluations and analysis. Finally, we conclude our work in Section 4.

## 2. Materials and Methods

The proposed method in this paper, shown in Figure 3, is mainly composed of the global Jacobian disentangled method and the local Jacobian disentangled method. In the global Jacobian disentangled method, the generator network uses the pre-trained StyleGAN2 model, our method first utilizes the weight matrix eigen decomposition method in the pre-trained network to obtain the original semantic attribute directions that are used as the initial direction set of the Jacobian orthogonal regularization; subsequently utilizes the loss function to train and update to obtain the final image editing direction. Compared with the common Jacobi orthogonal regularization method OroJaRGAN, we optimize the way of initialization during semantic direction search to improve the time efficiency and the proportion of effective semantic directions. The local Jacobi method, compared with OroJaRGAN, limits the search area of the image feature space, which utilizes a local contrast regularized loss function to relax the semantic association local area and non-local area and enables the local Jacobi disentangled method to obtain the local accurate semantic direction that cannot be searched by traditional methods and improve the editing effect of the generated image.

### 2.1. Global Jacobian Disentangled Method

The global Jacobi disentangled method firstly extracts the style layer weights of the generator network of the pre-trained StyleGAN2; secondly utilizes the eigen decomposition of the weight matrix and extracts the first k feature vectors as the initial set of semantic directions; finally, completes the Jacobi orthogonal regularization search and trains to update the set of semantic directions.

#### 2.1.1. Motivation of the Method

The weight matrix eigendecomposition method SefaGAN [20] first proposed that the parameters of the pre-trained GAN generator network are associated with the direction of image editing. The associated experiments show that the eigendecomposition of the tensor matrix can be performed in real-time, which has an advantage in time efficiency compared to the Jacobi orthogonal regularization search method. However, the method only utilizes the fully connected layer of the style layer, ignoring the mapping function of the other layers of the StyleGAN2 generator network. Therefore, when calculating the control variables of the latent space input to the output feature map, the semantic attribute direction is not accurate enough and there is still an entangled phenomenon [21].

Jacobi orthogonal regularization search algorithm [28] exploits the gradient relationship between the input and output feature maps of the generator network rather than the parameter information of a particular layer, which ensures the integrity of the method on the model. Jacobian Decomposition GAN [29] discovered and demonstrated that the eigenvectors obtained by the Jacobian matrix decomposition of the feature maps generated by the GAN generator network correspond to the semantic attribute direction of the image. OroJaRGAN introduced the concept of the Jacobi matrix in unsupervised image editing, expected that the biased derivatives of the output to the elements of each dimension of the input are mutually orthogonal, and proposed a training method of Jacobi orthogonal regularization to find the semantic attribute direction. The Jacobian orthogonal regularization search method utilizes the global mapping function of the GAN network from input to output, so the accuracy of the semantic attribute direction obtained is better than that of the weight matrix eigendecomposition method [30]. However, due to the random initialization of the search direction vector, the training Loss function converges slowly and the proportion of ineffective semantic directions is high.

To verify the effectiveness of the vector initialization by the Jacobi orthogonal regularization method, we counted the loss function curves and effective search vector statistics curves for the initialization direction of the OroJaRGAN method. As shown in Figure 4a, the Loss function curves are trained by the random initialization methods of Eyematrix_Init and Kaiming_normal_Init [31], respectively. We utilize a pre-trained StyGAN2 network on the FFHQ dataset, the images are down-sampled to 512 × 512 resolution, training parameters by default, and two GeForce RTX 2080Ti graphics cards are used for training in parallel. It can be seen that when the Loss function converges, the training time is not less than five hours. Figure 4b is the search effective direction statistics, the total number of vectors preset for each search is 40, where the vertical coordinate is the number of effective directions, the horizontal coordinate is the number of experiments, a total of ten sets of experiments, it can be seen that the highest proportion of the third and tenth experiments 9/40 = 21.5%, the search efficiency is low while wasting a lot of time resources.

#### 2.1.2. Method Feasibility Validation

Rather than optimizing the semantic direction search process from the perspective of accelerating the Jacobi matrix computation, we are exploring the impact of the initialization method on the time efficiency of the search process from another perspective, which is similar to pre-training in deep learning. In a single dataset, the direction vectors governing similar semantic attributes should be the same. Considering the consistency of the objectives of the weight matrix eigen decomposition method and the Jacobian orthogonal regularization search method in obtaining semantic directions and the complementarity in the efficiency and accuracy of the search, we utilize the direction vector set quickly obtained by the weight matrix eigen decomposition method as the initial Jacobian orthogonal regularization search vector set, and build a hybrid global disentangled method structure.

This structure is feasible when the direction vectors of the two methods are similar. We utilize cosine similarity to measure the correlation of the same semantic attribute direction of the two methods [32] as
(1)$$\mathrm{Similarity}=\cos\theta ({\vec{n}}_{1},{\vec{n}}_{2})=\frac{{\vec{n}}_{1}\cdot {\vec{n}}_{2}}{\left\|{\vec{n}}_{1}\right\|\left\|{\vec{n}}_{2}\right\|}=\frac{\sum_{i=1}^{n}\left(x_{i}*y_{i}\right)}{\sqrt{\sum_{i=1}^{n}{\left(x_{i}\right)}^{2}}*\sqrt{\sum_{i=1}^{n}{\left(y_{i}\right)}^{2}}},\;$$
where ${\vec{n}}_{1}=\left[x_{1},x_{2},x_{3},\ldots ,x_{n}\right]$, ${\vec{n}}_{2}=\left[y_{1},y_{2},y_{3},\ldots ,y_{n}\right]$ represent the same semantic attribute direction vector obtained by the weight eigen decomposition method and the Jacobi orthogonal regularization method, respectively. On the human face dataset FFHQ, the degree of similarity between the two methods on the direction vectors of the pose, age, hairstyle, and face color is counted for a total of ten sets of experiments, and the results are taken as the mean values, as shown in Table 1.

Table 1 suggests that on the five semantic attributes in the FFHQ dataset, the cosine similarity of the direction vectors obtained by both methods is greater than 0.80, which indicates that they have similar effectiveness, meanwhile, it is completely feasible to use the semantic direction of the weight matrix eigen decomposition as the initialization direction of the Jacobi orthogonal regularization method.

#### 2.1.3. Principle of Global Jacobi Disentangled Method

The semantic attribute directions are initialized by SefaGAN [20] by using the weight matrix eigendecomposition method as
(2)N*=argmaxN∈ℝd×k∑i=1k‖An→i‖22−∑i=1kλi(n→iTn→i−1),
where N=[n→1,n→2,⋯,n→k] is the set of the first k semantic attribute directions at each scale; n→ is the unit direction vector of semantic attributes; A is the matrix of all style layer weights stitched together; λ is the eigenvalue. The optimal solution as
(3)2ATAnj−2λjnj=0.

Each semantic attribute direction nj is an eigenvector of the weight matrix ATA, and we extract the first k eigenvectors to build the matrix Q and assign it to the initial matrix DInit∈ℝm×N as
(4)DInit=[Q1,m,Q2,m,⋯,Qk,m],
where m is the dimensionality of the latent space, and the column vector of DInit corresponds to a certain semantic attribute direction.

The set of optimal semantic attribute directions $D^{*}$ is finally obtained by iterative optimization by minimizing the expectation of the loss function as
(5)$$D^{*}=\arg\min_{D}E_{z,\omega_{i}}\left[{ℒ}_{J}\left(G\left(z+\eta D_{Init}\omega_{i}\right)\right)\right],$$
where G is the generator network; η represents the step size for moving along the semantic attribute direction; $\omega_{i}\in {\left\{0,1\right\}}^{N}$ as the column vector of a one-hot vector [33] index D. As with OroJaRGAN, the Jacobi orthogonal regularized loss function as
(6)$${ℒ}_{J}(G)=\sum_{d=1}^{K}\left\|J_{d}^{T}J_{d}{}^{\circ}(1-I)\right\|=\sum_{d=1}^{K}\sum_{i=1}^{m}\sum_{j\neq i}{\left|{\left[\frac{\partial G_{d}}{\partial z_{i}}\right]}^{T}\frac{\partial G_{d}}{\partial z_{j}}\right|}^{2},$$
where $G_{d}$ represents the feature map output by the *d*-th layer in the GAN network; $z=\left[z_{1},z_{2},z_{3,\cdots },z_{m}\right]$ represents the original latent space; $z_{i}$ represents the *i*-th dimension; $Jd=\left[j_{d,1},\cdots ,j_{d,i},j_{d,m}\right]$ represents the Jacobian matrix of $G_{d}$ relative to z. The operator ∘ represents the Hadamard product; I represents the identity matrix; 1 represents a matrix whose elements are all 1; $j_{d,i}=\frac{\partial G_{d}}{\partial z_{i}}$ represents the Jacobian vector.

The objective of the ${ℒ}_{J}\left(G\right)$ loss function is to train this loss function so that each dimension of the latent space z of the GAN network input controls each feature change of the image. The changes are caused by two different latent dimensions $z_{i}$ and $z_{j}$ should be independent. A prototype of this loss function is obtained by imposing this regularization restriction on all dimensions of z, $z_{i}$, i∈[1,m] and all layer outputs of the GAN network, $G_{d}$, d∈[1,k]. For a pre-trained GAN network, we are actually looking for a direction vector to move along z, perturbing one or more dimensions to change a particular semantic property.

Figure 5 compares random initialization and the initialization by the eigenvectors in this paper in terms of the convergence time of the loss function and the proportion of effective vectors. In Figure 5a, the Jacobi orthogonal regularization search algorithm based on the weight matrix eigendecomposition improves the search efficiency by reducing the training time from 5.5 h to 2.5 h than the Jacobi orthogonal regularization search algorithm. In Figure 5b, the eigenvector initialized Jacobi search algorithm has an average effective direction of 18 and a maximum value of 20 among 40 initialization direction vectors, which is double the proportion of effective directions of random initialization. The experiments demonstrate the effectiveness of the global Jacobi disentangled method.

### 2.2. Local Jacobian Disentangled Method

#### 2.2.1. Local Jacobi Orthogonal Regularization Search Algorithm

The global Jacobi disentangled method has good performance on the global image generated by the StyleGAN2 network, but it is difficult to search the semantic attribute direction in local areas such as glasses, mouth, nose, hair, etc., and there is still serious entanglement with other semantic attributes when editing images. As shown in Figure 6, when the reference image in the red box changes along the semantic direction of the face glasses, the semantic attributes such as age also change. This is because the spatial distribution of local semantic features is concentrated and mainly distributed in high-resolution images when there is a more complex entangled relationship between global and local semantics, and the global Jacobi method is difficult to realize a complete decoupling [34].

We observe that the semantic attributes of most images in the FFHQ dataset trained by the StyleGAN2 network are relatively fixed in position on the feature space. Inspired by the local area mask idea of MaskGAN [35], Editing in Style [36], we proposed a local Jacobi orthogonal regularization search method based on a priori rectangular mask. Compared with the global Jacobi disentangled algorithm, the local Jacobi disentangled algorithm limits the feature map area searched at each resolution layer; divides the feature map of each resolution layer of the generator into two parts: local In-area and non-local Out-area; finally completes the local semantic disentanglement for the local feature map according to the Jacobi disentangled method in Equations (2)–(6). Figure 7 shows the mouth, eye, and hair areas of a human face. We define the red box as the local area of relevant semantic attributes and the non-local area outside the red box.

To verify the effectiveness of the local Jacobi disentangled method proposed in this paper, for the FFHQ dataset, the eye area is mainly selected for experiments, and the area of Jacobi orthogonal regularization search is limited. The final obtained example of the image editing direction of the semantic attributes of glasses is shown in Figure 8, where the semantic attributes of the eyes are significantly changed within the limited search area. However, after using the local area search method, the hair color and character ID outside the local area changed significantly during the semantic editing of glasses. This is because local Jacobi search obtains local semantics, while image editing moves along the direction of global semantics and does not limit the semantic editing area.

#### 2.2.2. Local Contrast Regularized Loss Function

To solve the problem of semantic attribute entanglement of local Jacobi image editing, we refer to LELSD [37], IndomainGAN [38], and optimization strategies [39] to propose a local contrast regularized loss function to relax the semantic tight entangled relationship local and non-local area. Local contrast regularization means that the semantic direction should satisfy both constraints of maximum variation of image features inside the area and minimum variation of image features outside the area, as
(7)$$Diff_{\mathrm{In}-\mathrm{area}\;}=\max_{D}G\left(z+\eta D\omega_{i}\right)-G(z),$$
(8)$$Diff_{\mathrm{Out}-\mathrm{area}\;}=\min_{D}G\left(z+\eta D\omega_{i}\right)-G(z),$$
where G(z) represents the original generated image, and $G\left(z+\eta D\omega_{i}\right)$ represents the image after editing along the semantic attribute direction.

We fused the two constraints to propose a local contrast regularized loss function as
(9)$${ℒ}_{In-Out}=T\left(G_{\mathrm{Out}-\mathrm{area}\;}\left(z+\eta D\omega_{i}\right)-G(z)\right)-K\left(G_{\mathrm{In}-\mathrm{area}\;}\left(z+\eta D\omega_{i}\right)-G(z)\right),$$
where K and T are the balance parameters during training, which are used to stabilize the training process. The final loss function is as
(10)$$Loss={ℒ}_{J}+{ℒ}_{In-Out}.$$

The optimal semantic attribute vector matrix $D^{*}$ is
(11)$$D^{*}=\arg\min_{D}\;E_{z,\omega_{i}}\left[{ℒ}_{J}\left(G_{\mathrm{In}-\mathrm{area}\;}\left(z+\eta D_{Init}\omega_{i}\right)\right)+{ℒ}_{In-Out}\left(G\left(z+\eta D_{Init}\omega_{i}\right)\right)\right].$$

The results of semantic editing after re-training by applying the new optimization function are shown in Figure 9. The semantics inside the eye area is obviously changed, and the outside of the eye area is almost unchanged, which is a significant improvement compared with Figure 8 and achieves a better local semantic disentanglement.

## 3. Results

### 3.1. Experimental Details

**Datasets.** In this paper, we use the mainstream face dataset FFHQ and the cat dataset LSUNCat, both of which are the primary evaluation dataset for most generated image attribute edits methods. Among them, the FFHQ dataset has a large number of images and a pure background of face images. Due to the performance limitation of the computing platform, the FFHQ face dataset is down-sampled from 1024 × 1024 resolution to 512 × 512 resolution with 70 K images. the LSUNCat dataset has 256 × 256 resolution with the same 70 K images.**Parameter setting.** The total number of iterations for Jacobi orthogonal regularization training is 5 × 10^4^ for the FFHQ dataset and 4 × 10^4^ for the LSUNCat dataset. The number of column vectors of the initialization matrix D in the direction of semantic attributes is 40, and the local and non-local regularization training balance parameters K, T are taken as 0.6 and 0.4, respectively.**Experimental environment.** The code is executed on Ubuntu 18.04 with Intel(R) Core(TM) i7-7820X CPU @ 3.60GHz and GeForce RTX 2080Ti×2. The deep learning framework is Pytorch.**Evaluation Metrics.** The perceptual path distance (PPL) [40] is used as a metric to measure the performance of semantic attribute disentanglement. This metric describes how drastically the image changes when the intermediate latent space w is interpolated along a certain direction, and its small value represents a relatively smooth latent space and low entanglement. Referring to STIA-WO [22] the PPL value is calculated for the intermediate latent space w with a certain range of a sampling point along its orthogonal semantic attribute direction, instead of randomly sampling two latent spaces w for calculating the PPL value as
(12)$$PPL=E\left[\frac{1}{\epsilon^{2}}d\left(G\left(w\right),G\left(w+\epsilon \cdot n\right)\right)\right],$$where ε = 10^−4^ represents the range moved during editing, d(⋅,⋅) is the perceptual distance between the two generated images [41], and G represents the generator. The two sampling points corresponding to the images are the intermediate latent space w and its points along the unit attribute direction $\vec{n}$ shift ε.

### 3.2. Experimental Results Comparison

We utilize PPL to evaluate the performance of the global Jacobi disentangled method compared with the three mainstream unsupervised generated image editing methods, SefaGAN, PCAGAN, and OroJaRGAN, on the FFHQ dataset. The comparison results shown in Table 2 suggest that the performance of our method and the OroJaRGAN method is close and slightly better than the other two methods, but the training efficiency and search efficiency of our method are significantly improved compared with OroJaRGAN. Figure 10 and Figure 11 show the comparison of the four editing methods for two semantic attributes, gender, and age, on the FFHQ face dataset. In two examples, the PCAGAN method has entanglement with age attribute on gender attribute editing and more serious entanglement with hair attribute on age attribute editing; SefaGAN has an insignificant effect on gender attribute editing and more serious entanglement with pose occurs on age attribute editing. In contrast, the OroJaRGAN method and the proposed method obtained more reasonable results.

We evaluated four methods on the LSUNCat dataset in the same way, and the results are shown in Table 3. The disentangled performance of this method is similar to OroJaRGAN and better than PCAGAN and SefaGAN. We compared the results of semantic attribute editing for rotation and cat coat color, and the results are shown in Figure 12 and Figure 13. In both examples, PCAGAN and SefaGAN are more severely entangled with the cat poses in rotation semantic attributes, and more severely entangled with the global color in cat coat color, compared with these two methods in this paper. The editing effect of our method is significantly improved.

Our method is similar to OroJaRGAN in terms of PPL metrics because the search method is the same, only the semantic direction is initialized in a different way. Referring to papers [42,43], the equivalent model of SefaGAN considered only that the role of the mapping function in the network is limited to the style layer, ignoring the role of other layers. the effect of PCAGAN depends heavily on the number of sampling points. In contrast, our method resembles an end-to-end form when computing the Jacobi matrix without the problems of the two methods above. So, our method has an improvement in the PPL metric.

To verify the effectiveness of the local Jacobian disentangled method, we compared the performance of the local contrast regularization method with the non-regularization method in local semantic attribute disentanglement, and the PPL score is still used as the evaluation metric. Four representative local semantic attributes, glasses, mouth, hairstyle, and hair color, were selected for the FFHQ dataset, and three local semantic attributes, i.e., head posture, head color, and abdominal color, were selected for the LSUNCat dataset. The results are shown in Table 4 and Table 5.

As can be seen from Table 4 and Table 5, in both datasets, the semantic attributes obtained by the non-regularization disentangled method are more severely entangled, because the other semantics change when the semantics is edited, and the controllability is poor. After imposing the contrast regularization constraint, the association of the four semantics in the FFHQ dataset was significantly reduced, and the average PPL decreased by 90%, among which the mouth semantics and the glasses semantics achieved 94% and 91% association reduction, respectively. In the LSUNCat dataset, the average PPL decreased by 93%, among which the abdominal color semantics achieved a 95% association decrease, indicating that the local contrast regularization achieved effective disentanglement of local semantic attributes and found accurate semantic directions, which could not be achieved by the other methods.

Figure 14 and Figure 15 show the image editing results with local contrast regularization in the two datasets. It can be seen that the semantics of glasses and mouth shape are basically decoupled in the face dataset, and the person ID maintains a good consistency. However, there is still a certain degree of coupling when editing the semantics of hair and gender, because gender is a global semantics and hair is a local semantics, and the entanglement between global and local semantics is complicated. The local contrast regularization can relax the relatively independent local semantics, but its effect on decoupling global and local semantics is not obvious. Additionally, the accurate direction of the semantic properties of hair color is more meaningful. Similarly, the accurate head poses semantic attribute direction can be found in the LSUNCat dataset, and the head color and abdominal color are essentially disentangled from the color of the non-local area.

## 4. Conclusions

In this paper, we proposed a generated image editing method based on global-local Jacobi disentanglement. Our global Jacobi disentangled method can improve the time efficiency of searching semantic directions with the proportion of valid directions, which makes the method deployable on platforms with less computational power. The local Jacobi disentangled method can obtain more and more accurate semantic directions locally, improving the accuracy and reliability of image editing. However, the method in this paper still has limitations to improve the training efficiency by improving the initialization method, it is required to research an algorithm to speed up the computation of the Jacobi matrix, and our method should be applied to other GAN models in addition to the StyleGAN2 model. The next step will be to carry out more extensive research to enhance the applicability of our approach.

## Figures and Tables

**Figure 1 sensors-23-01815-f001:**
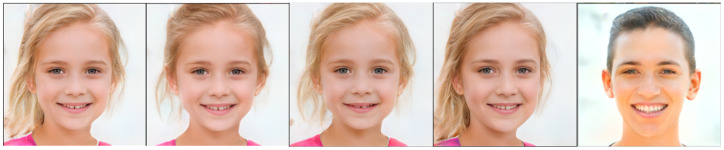
Generated human face image editing on semantic attributes of hairstyle, age, pose, and gender respectively.

**Figure 2 sensors-23-01815-f002:**
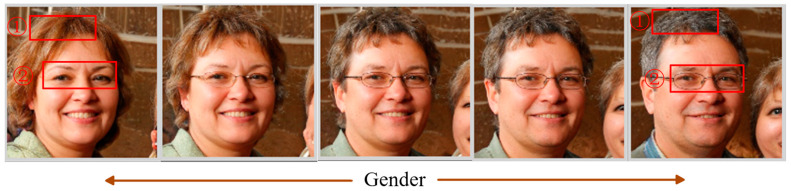
Entanglement of gender attributes with other semantic attributes in SefaGAN.

**Figure 3 sensors-23-01815-f003:**
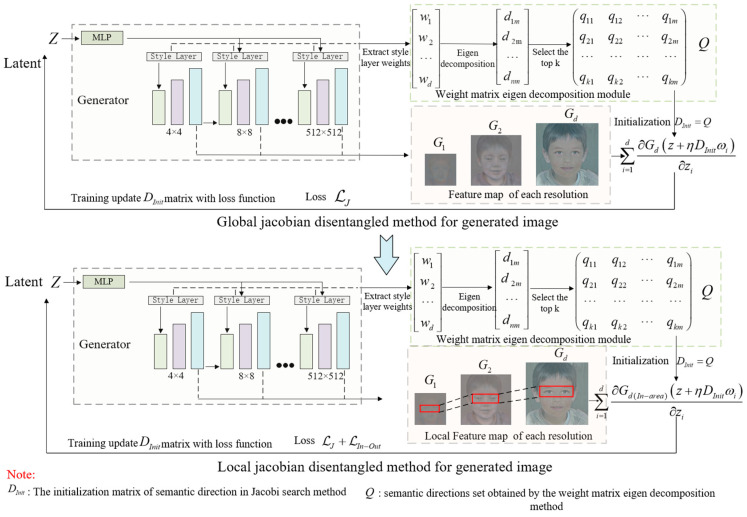
The overall architecture of our proposed generated image editing method.

**Figure 4 sensors-23-01815-f004:**
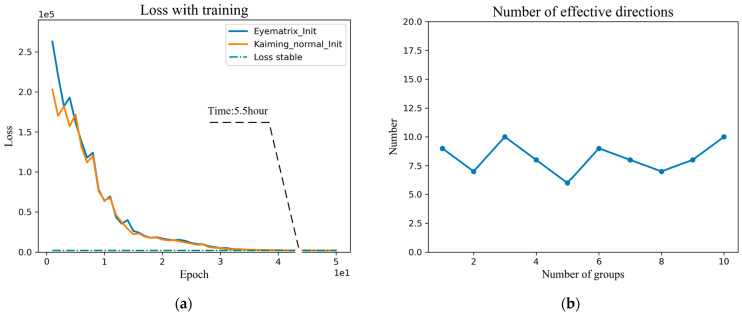
(**a**) Training time of Jacobi orthogonal regularization search with different initialization methods. (**b**) The proportion of effective directions obtained by Jacobi orthogonal regularization search method.

**Figure 5 sensors-23-01815-f005:**
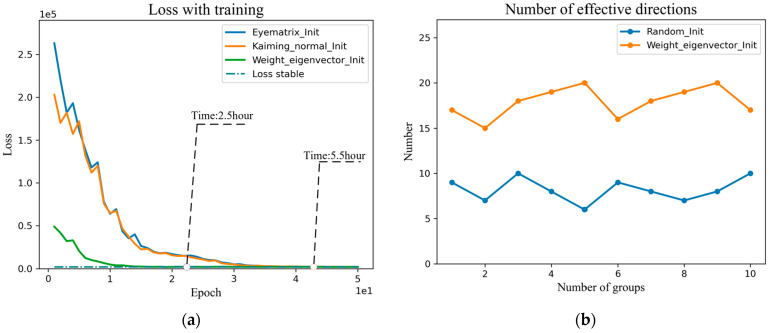
(**a**) Comparison of training time for loss convergence. (**b**) Comparison of the proportion of effective semantic editing directions.

**Figure 6 sensors-23-01815-f006:**
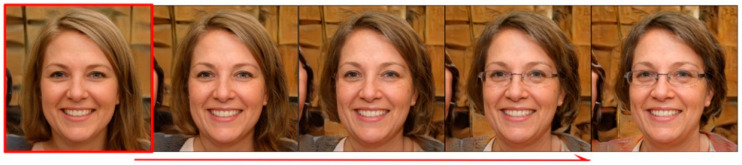
Entanglement of glasses attributes and age attributes in face editing.

**Figure 7 sensors-23-01815-f007:**
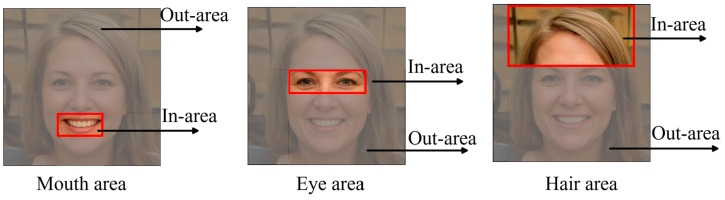
Semantic attribute local and non-local area examples of face mouth, eye, and hair areas.

**Figure 8 sensors-23-01815-f008:**
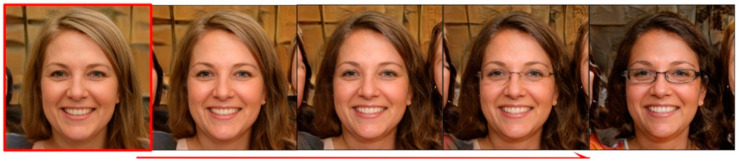
The result of editing the semantic attribute of glasses without local contrast regularization.

**Figure 9 sensors-23-01815-f009:**
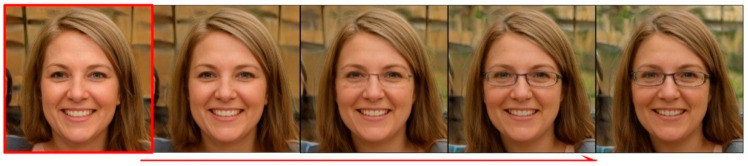
The result of editing the semantic attribute of glasses with local contrast regularization.

**Figure 10 sensors-23-01815-f010:**
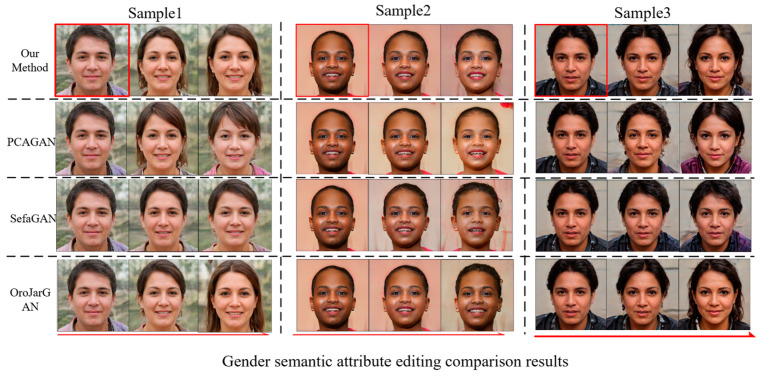
FFHQ dataset faces gender semantic attribute editing comparison results.

**Figure 11 sensors-23-01815-f011:**
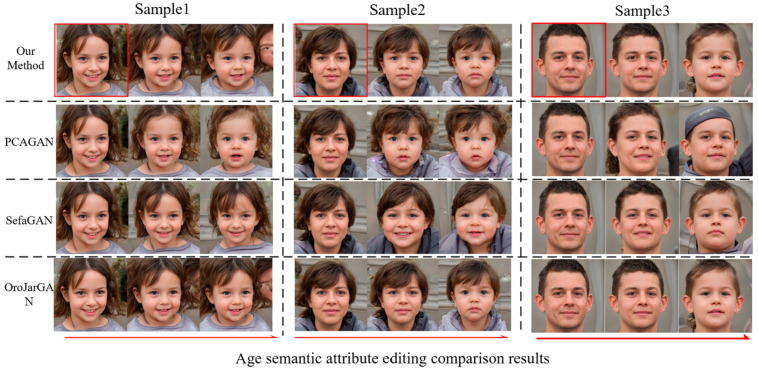
FFHQ dataset faces age semantic attribute editing comparison results.

**Figure 12 sensors-23-01815-f012:**
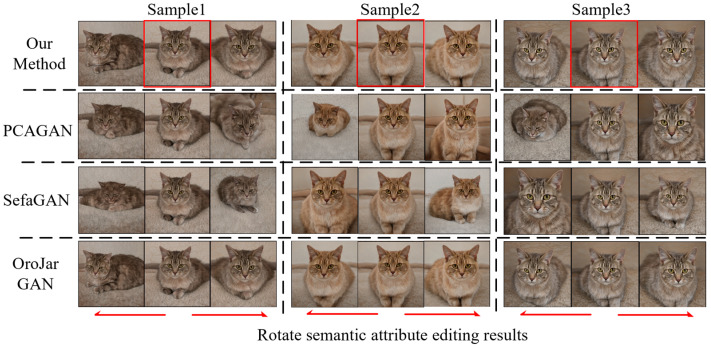
LSUNCat dataset rotates semantic attribute editing comparison results.

**Figure 13 sensors-23-01815-f013:**
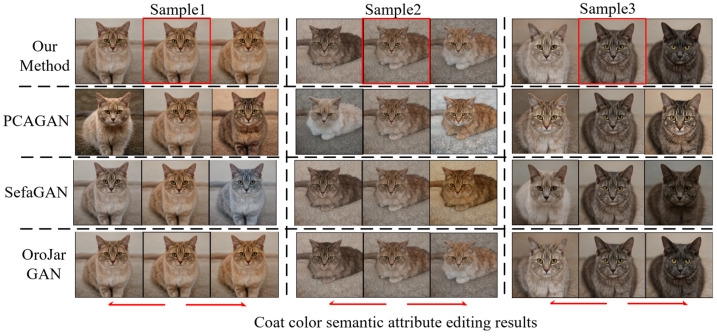
LSUNCat dataset Coat color semantic attribute editing comparison results.

**Figure 14 sensors-23-01815-f014:**
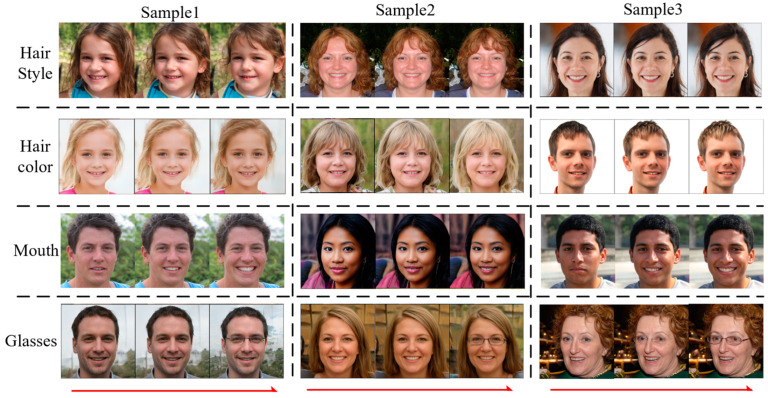
Local Jacobi disentangled method editing results for the FFHQ dataset.

**Figure 15 sensors-23-01815-f015:**
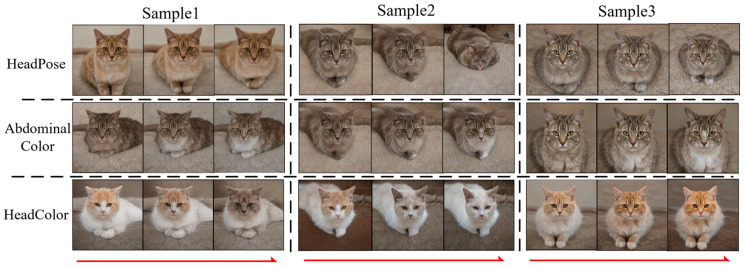
Local Jacobi disentangled method editing results for the LSUNCat dataset.

**Table 1 sensors-23-01815-t001:** Semantic attribute editing direction cosine similarity.

	Pose	Age	Gender	Hairstyle	Face Color
**Cosine Similarity**	0.91	0.89	0.86	0.87	0.84

**Table 2 sensors-23-01815-t002:** PPL Scores of FFHQ dataset semantic attribute directions (↓).

Model	Pose	Age	Gender	Hairstyle	Face Color
**SefaGAN**	0.84	1.01	0.98	0.94	0.90
**PCAGAN**	0.76	0.96	0.92	0.87	0.89
**OroJaRGAN**	0.70	**0.85**	0.84	0.80	0.84
**Our Method**	**0.69**	0.87	**0.82**	**0.76**	**0.77**

**Table 3 sensors-23-01815-t003:** PPL scores of LSUNCat dataset semantic attribute directions (↓).

Model	Rotate	Scale	Coat Color
**SefaGAN**	0.82	0.73	0.71
**PCAGAN**	0.76	0.68	0.65
**OroJaRGAN**	0.57	0.49	0.41
**Our Method**	**0.56**	**0.47**	**0.40**

**Table 4 sensors-23-01815-t004:** PPL scores of FFHQ dataset local Jacobi disentangled method attribute directions (↓).

Model	Glasses	Mouth	Hairstyle	Hair Color	Average
**Non-regularization**	8.92	12.56	3.12	4.89	7.37
**Local contrast regularization**	**0.83**	**0.72**	**0.76**	**0.69**	**0.75**

**Table 5 sensors-23-01815-t005:** PPL scores of LSUNCat dataset local Jacobi disentangled method attribute directions (↓).

Model	Head Pose	Head Color	Abdominal Color	Average
**Non-regularization**	6.46	8.56	9.12	8.05
**Local contrast regularization**	**0.43**	**0.52**	**0.46**	**0.52**

## Data Availability

Not applicable.

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
