# Peer review of "Generated Image Editing Method Based on Global-Local Jacobi Disentanglement for Machine Learning"

_sensors, 2023, doi:10.3390/s23041815_

Round 1

Reviewer 1 Report

In this manuscript, a generated image editing method based on hierarchical Jacobi disentanglement was developed. By comparison with other methods, the method proposed can improve the efficiency of global semantic search and discover more accurate local semantic attributes. It is meaningful and interesting. Therefore, the reviewer thinks that this paper would be suitable for publication in Sensors. However, in the reviewer’s opinion, minor clarifications are needed before the publication.

1) Whether the sample is sufficient to support your analyzed results.

2) The PPL is used for evaluation of the method. Indeed, from the results in Table2, 3 and so on, the editing performance was improved by the developed method, but the reason why it can improve the editing effect was not explained clearly. Please add some relative theoretical explanation in the revised manuscript.

3) The quality of Fig. 4 and 5 is not high, please modify figures.

4) There are several spelling mistakes in this manuscript.  Please check and correct carefully.

Reviewer 2 Report

please see the attachment for details

Reviewer 3 Report

The authors in this paper propose a generated image editing method based on hierarchical Jacobi disentanglement where the global search utilizes the weight matrix eigen vectors as the  initial vectors of Jacobi orthogonal regularization search method. The approach used in this paper is double the search effectiveness as well as significantly improves the training convergence speed and highly help in discovering more accurate local semantic attributes.

The problem is clear from the title. The authors need to describe the method clearly and then explain the novelty of their study in a detailed paragraph. Specifically, the innovation compared to related studies must be highlighted and discussed.

While the paper does a good job of putting the hierarchical Jacobi disentanglement in the context of generative adversarial networks. But the problem of high computational complexity of Jacobi matrix is not sufficiently discussed.

Add the limitation of the study to the conclusion section

 What are the practical implications of this study?

The related work needs to add. Specifically, a critical discussion on state of the art is missing here. Please revise

Round 2

Reviewer 2 Report

Thanks for the authors' detailed replies. The questions have been well addressed.

Reviewer 3 Report

thanks for your works